# Evaluation of the Antioxidant, Antidiabetic, and Antiplasmodial Activities of Xanthones Isolated from *Garcinia forbesii* and Their *In Silico* Studies

**DOI:** 10.3390/biomedicines9101380

**Published:** 2021-10-02

**Authors:** Johanis Wairata, Edwin Risky Sukandar, Arif Fadlan, Adi Setyo Purnomo, Muhammad Taher, Taslim Ersam

**Affiliations:** 1Department of Chemistry, Faculty of Science and Data Analytics, Institut Teknologi Sepuluh Nopember (ITS), Kampus ITS Sukolilo, Surabaya 60111, Indonesia; jhoeuniera@gmail.com (J.W.); edwin.risky.s@gmail.com (E.R.S.); afadlan@chem.its.ac.id (A.F.); adi.spurnomo@yahoo.com (A.S.P.); 2Department of Agrotechnology, Faculty of Natural Science and Engineering Technology, University Halmahera, Jalan Wari-Ino, Tobelo, North Halmahera 97762, Indonesia; 3Department of Pharmaceutical Technology, Kulliyyah of Pharmacy, International Islamic University Malaysia, Bandar Indera Mahkota, Kuantan 25200, Malaysia; mtaher@iium.edu.my

**Keywords:** *Garcinia forbesii*, xanthones, antioxidant, antidiabetic, antiplasmodial, molecular docking

## Abstract

This study aimed to isolate xanthones from *Garcinia forbesii* and evaluated their activity in vitro and *in silico*. The isolated compounds were evaluated for their antioxidant activity by DPPH, ABTS and FRAP methods. The antidiabetic activity was performed against α-glucosidase and α-amylase enzymes. The antiplasmodial activity was evaluated using *Plasmodium falciparum* strain 3D7 sensitive to chloroquine. Molecular docking analysis on the human lysosomal acid-alpha-glucosidase enzyme (5NN8) and *P. falciparum* lactate dehydrogenase enzyme (1CET) and prediction of ADMET for the active compound, were also studied. For the first time, lichexanthone (**1**), subelliptenone H (**2**), 12b-hydroxy-des-D-garcigerrin A (**3**), garciniaxanthone B (**4**) and garcigerin A (**5**) were isolated from the CH_2_Cl_2_ extract of the stem bark of *G. forbesii*. Four xanthones (Compounds **2**–**5**) showed strong antioxidant activity. In vitro α-glucosidase test showed that Compounds **2** and **5** were more active than the others, while Compound **4** was the strongest against α-amylase enzymes. In vitro antiplasmodial evaluation revealed that Compounds **2** and **3** showed inhibitory activity on *P. falciparum*. Molecular docking studies confirmed in vitro activity. ADMET predictions suggested that Compounds **1**–**5** were potential candidates for oral drugs. The isolated **2**–**5** can be used as promising phytotherapy in antidiabetic and antiplasmodial treatment.

## 1. Introduction

The genus *Garcinia* belongs to the family Clusiaceae of subfamily Clusioideae and tribe Garcinieae with more than 600 species are widely distributed in Africa, Asia, Australia, Polynesia, and Latin America [1]. *Garcinia* is mainly found in tropical rain forest of Southeast Asia and West Africa [2] and rich of secondary metabolites i.e., biflavonoids, xanthones, biphenyls, polyprenylated benzophenone, depsidones, and triterpenoids with various bioactivities. The aqueous extract of *Garcinia combogia* and ethyl acetate extract of *Garcinia xanthochymus* showed antioxidant activity [3,4]. Further, 6-*O*-methyl-2-deprenylrheediaxanthone B, vieillardixanthone, forbexanthone, buchanaxanthone, isocudraniaxanthone A, and 5,7-dihydroxychromone have been isolated from the stem bark of *Garcinia vieillardii* and strongly active on DPPH assay [5]. Furthermore, polyisoprenylated xanthone guttiferone A from *Garcinia aristata* fruit displayed antiplasmodial activity against *P**lasmodium*
*falciparum* with IC_50_ values of 0.5 μM, comparable to chloroquine (IC_50_ values of 0.3 μM) [6]. Mckeanianones A–E and new biflavone mckeaniabiflavone isolated from *Garcinia mckeaniana* had been reported active to *P. falciparum* strain TM4 and K1 [7]. Moreover, norcowanin isolated from the ethyl acetate extract of *Garcinia oblongifolia* twig actively inhibited α-glucosidase enzyme with IC_50_ values of 1.7 μM [8]. Fractination of ethyl acetate extract of *Garcinia xanthochymus* stem bark yielded xanthochymusxanthones A and B which showed significant activity toward α-glucosidase [9].

*Garcinia forbesii* King is a member of *Garcinia* genus native to Sumatera and Borneo islands of Indonesia. This species is popular as forest red mangosteen in local community with sweet-sour taste of smaller size fruit than common mangosteen fruit (*Garcinia mangostana* L.). The peel of *G. forbesii* King fruit is commonly used as herbs and spices and provides sour taste in various types of dishes [10]. To the best of our knowledge, there is only one report on the phytochemicals of this species and there is no bioactivity evaluation of this plant so far. In 1993, Harrison [11] and co-workers reported the isolation of forbexanthone, piranojacareubin, and 1,3,7-trihydroxy-2-(3-methylbut-2-enyl)-xanthone from *G. forbesii* King. In this study, the extraction and isolation of active compounds from the stem bark of *G. forbesii* King were reported. The isolated compounds were then assayed for their antioxidant potential using 2,2-Diphenyl-2-picrylhydrazyl (DPPH), 2,2-azinobis-3-ethylbenzothia-zoline-6-sulfonic acid (ABTS), and ferric reducing-antioxidant power (FRAP) methods along with in vitro antidiabetic assay (α-glucosidase and α-amylase inhibitory activities) and in vitro antiplasmodial activity against *P. falciparum* strain 3D7 sensitive to chloroquine. The *in silico* studies were performed by molecular docking and absorption distribution metabolism excretion toxicity (ADMET) prediction to gain a deeper understanding of the correlation between the activities and oral drugs potency.

## 2. Materials and Methods

### 2.1. General Experiment Procedures

Vacuum liquid chromatography (VLC) and column chromatography (CC) (silica gel 60, 63–200 μm; Merck), Sephadex LH-20 (25–100 μm; GE Healthcare), and TLC analysis (silica gel 60 GF254, 0.25 mm; Merck) were utilized for isolation of the compounds. Melting point was determined with Fischer–Johns melting point Apparatus (Philip Haris, Calgary, Canada). IR data were obtained using an FT-IR Spectrophotometer (Shimadzu 8400S, Kyoto, Japan). A UV-Vis Spectrophotometer (Genesys, Thermo Fisher Scientific, Madison, WI, USA) was used for UV-Vis spectra collection. The NMR spectra (^1^H 400 MHz and ^13^C 100 MHz) were recorded on NMR spectrometer (JEOL ECS, Tokyo, Japan) using the appropriate deuterated solvents. The HRESIMS spectra were collected by using a mass spectrometer (Water Xevo Q-tof MS, Milford, MA, USA). Docking study was performed with the Toshiba Portege Computer tool, Intel^®^ Core ™ i7-6600U CPU @ 2.60GHZ 2.80GHZ, 8.00 GB RAM, Intel HD Graphics 520. Molecular docking was studied by Molegro Virtual Docker 5.5. The structure of ligands was drawn by using ChemDraw 2018, minimized by Chem3D 2018, and saved as. mol2 format files.

### 2.2. Plant Material

The stem barks of *G. forbesii* King were collected from the Somaetek forest in North Halmahera, Indonesia (1°25′22″ LU 127°46′59″ E). The specimen was stored in Bogor Botanical Gardens (specimen number VII.G.237a).

### 2.3. Extraction and Isolation

The dried bark of *G. forbesii* (3.5 kg) was ground into powder and macerated at room temperature with CH_2_Cl_2_ (3 × 15 L) for three days. The solvent was evaporated under reduced pressure to obtain the extract (121 g). The separation was then carried out by VLC with silica gel (300.0 g) using 100% *v/v n*-hexane, 100% *v/v* methylene chloride, 100% *v/v* ethyl acetate, and 100% *v/v* methanol to yield four fractions: *n*-hexane (F1: 8.0 g); CH_2_Cl_2_ (F2: 46.0 g); EtOAc (F3: 43.0 g), and MeOH (F4: 8.0 g) fractions. The CH_2_Cl_2_ fraction was separated with VLC on silica gel (250.0 g) using *n*-hexane:CH_2_Cl_2_ gradient (100:0, 95:5, 90:10, 85:15, 75:25, 50:50, 0:100 *v/v*) to give five subfractions (M1-M5). The subfraction M1 (1.57 g) was chromatographed on Sephadex LH-20 (100.0 g) eluted with CH_2_Cl_2_:MeOH (1:1 *v/v*) to produce Compounds **1** (33.0 mg) and **2** (54.0 mg). Subfraction M2 (2.5 g) was separated with sephadex LH-20 (100.0 g) using CH_2_Cl_2_: MeOH (1:1 *v/v*) to give compound **4** (62.0 mg) and compound **5** (150.0 mg). Compound **3** (256.0 mg) was obtained from subfraction M3 (3.5 g) separated by repeated chromatography on Sephadex LH-20 (100 g) with CH_2_Cl_2_: MeOH (30:70 *v/v*).

#### 2.3.1. Lichexanthone (1)

Pale yellow powder; mp: 186–188 °C; UV (MeOH) λ_max_: 302, 232, and 224 nm; IR ν_max_ (KBr): 3446, 2918, 1639, 1608, 1452, 1280, and 1207 cm^−1^, for ^1^H (400 MHz, CDCl_3_) and ^13^C (100 MHz, CDCl_3_) spectroscopic data, see Table 1; and HRESIMS *m/z* 287.0943 [M+H]^+^ (calcd. for C_16_H_15_O_5_, 287.0919).

#### 2.3.2. Subelliptenone H (2)

Yellow powder; mp: 175–177 °C; UV (MeOH) λ_max_: 352, 294, and 208 nm; IR ν_max_ (KBr): 3441, 2974, 1641, 1585, 1471, 1288, and 1207 cm^−1^, for ^1^H (400 MHz, DMSO-*d_6_*) and ^13^C (100 MHz, DMSO-*d_6_*) spectroscopic data, see Table 1; and HRESIMS *m/z* 395.1506 [M+H]^+^ (calcd. for C_23_H_23_O_6_, 395.1495).

#### 2.3.3. 12b-Hydroxy-des-D-garcigerrin A (3)

Yellow powder; mp: 207–209 °C; UV (MeOH) λ_max_: 310, 246, and 206 nm; IR ν_max_ (KBr): 3325, 2964, 1639, 1583, 1462, 1234, and 1178 cm^−1^, for ^1^H (400 MHz, DMSO-*d_6_*) and ^13^C (100 MHz, DMSO-*d_6_*) spectroscopic data, see Table 1; and HRESIMS *m/z* 335.0920 [M+Na]^+^ (calcd. for C_18_H_18_O_5_Na, 335.0895).

#### 2.3.4. Garciniaxanthone B (4)

Yellow powder; mp: 167–169 °C; UV (MeOH) λ_max_: 342, 262, and 206 nm; IR ν_max_ (KBr): 3527, 2976, 1637, 1618, 1450, 1290, and 1182 cm^−1^, for ^1^H (400 MHz, DMSO-*d_6_*) and ^13^C (100 MHz, DMSO-*d_6_*) spectroscopic data, see Table 2; and HRESIMS *m/z* 379.1558 [M+H]^+^ (calcd. for C_23_H_23_O_5_, 379.1545).

#### 2.3.5. Garcigerin A (5)

Yellow powder; mp: 246–248 °C; UV (MeOH) λ_max_: 398, 314, 250 and 206 nm; IR ν_max_ (KBr): 3423, 2978, 1637, 1608, 1464, 1230, and 1165 cm^−1^, for ^1^H (400 MHz, DMSO-*d_6_*) and ^13^C (100 MHz, DMSO-*d_6_*) spectroscopic data, see Table 2; and HRESIMS *m/z* 435.1437 [M+Na]^+^ (calcd. For C_23_H_24_O_7_Na, 435.1420).

### 2.4. Antioxidant Assay

#### 2.4.1. DPPH Radical Scavenging Assay

DPPH (1,1-diphenyl-2-picrylhydrazyl) was used for the antioxidant test using quercetin and gallic acid as controls. The compounds were diluted at concentrations of 159.73, 79.87, 39.93, 19.97, 9.98 µg/mL. A total 1 mL of DPPH 6 × 10^−5^ M solution, complete with 33 µL of the compound, was incubated at 37 °C for 20 min. The experiment was carried out with three replications and the absorbance was observed at a wavelength of 517 nm. The radical inhibitory activity was calculated using the equation [(Ab-As)/Ab] × 100%.

#### 2.4.2. ABTS Radical Scavenging Assay

The antioxidant activity was also determined by free radical ABTS (2,2″-azinobis (3-ethyl benzothiazoline-6-sulfonic acid) according to the reported procedure [12]. In this test, 7 mM ABTS solution was prepared and mixed with a buffer solution of potassium peroxydisulfate, incubated for 16 h, and protected from light. A total of 10 μL of the sample and 1 mL of the ABTS was incubated for 4 min and measured at a wavelength of 734 nm. This experiment was performed triplo using quercetin and gallic acid as controls. The ABTS inhibitory activity was calculated by the equation [(Ab-As)/Ab] × 100%.

#### 2.4.3. Ferric Reducing-Antioxidant Power (FRAP) Assay

The FRAP assay was determined according to the described method [13] using 300 mM acetate buffer, 40 mM HCl, 10 mM (2,4,6 Tris(2-pyridyl)-s-triazine (TPTZ)), 20 mM FeCl_3_·6H_2_O. For standard curve comparison, a 10:1:1 (acetate buffer: TPTZ: H_2_O) and sample ratio 10:1:1 (acetate buffer: TPTZ: FeCl_3_·6H_2_O) were constructed. The standard curve was prepared using various concentrations of FeSO_4_·7H_2_O. The reaction mixture was incubated at 37 °C for 30 min followed by measurement of the absorbance at 593 nm. In this test, the reduction capacity of the tested compound calculated concerning the reaction signal given by the Fe^2+^ solution. The FRAP value was expressed as µM Fe^2+^/g ((Frap value of sample (µM) = abs (sample) × FRAP value of std (µM)/abs (std).

### 2.5. In Vitro Antidiabetic Assay

#### 2.5.1. Rat Intestinal α-glucosidase Inhibitory Activity

The inhibitory activity of rat intestinal α-glucosidase was determined according to previously reported method [14] with slight modifications. This procedure was performed by classifying four mixtures in different groups. The mixture (1) labelled “Blank of enzyme reaction” contained the mixture and DMSO (10 µL), sodium phosphate buffer (50 µL), glucose kit (80 µL), enzyme (20 µL) and substrate (20 µL maltose, sucrose 20 µL). A 10 µL of sample and samples with various concentrations of compounds was added to the well Plates (3) and (4). A 50 µL of sodium phosphate buffer (pH 6.9) was then added to well Plates (1) and (3), and a 30 µL of the buffer was added to well Plates (2) and (4), respectively. The substrate (maltose, 20 µL; sucrose 20 µL) was then added to the Plates (2) and (4). A glucose kit (80 µL each) and the enzyme (20 µL) were added to all plates and were incubated for 10 min for maltose and 40 min for sucrose. Finally, the absorbance was then measured at λ = 520 nm using a microplate reader (BioTek ELx800TM, BioTek Instruments, Inc., Winooski, VT, USA).

#### 2.5.2. α-Amylase Inhibitory Activity

The modified α-amylase enzyme inhibition test was conducted for antidiabetic activity evaluation [15]. A total of 10 mg sample was dissolved in 1 mL DMSO and 0.1 M phosphate buffer and 5 mg α-amylase solution (Porcine pancreatic α-amylase) in 1 mL phosphate buffer pH 6.9 was added. A 100 mg of potato starch (substrate) was heated in 5 mL of 0.1 M phosphate buffer for 5 min and cooled to room temperature. Then, 20 µL of the sample and 50 µL of substrate were mixed into 30 µL of phosphate buffer. The mixture was pre-incubated for 5 min followed by the addition of 20 µL of α-amylase enzyme and then incubation at 37 °C for 15 min. The color development was performed by adding 50 µL of 1M HCl and 50 µL of 10% iodine solution. Finally, the absorbance was measured at a wavelength of 650 nm.

### 2.6. In Vitro Antiplasmodial Assay

The antiplasmodial activity was carried out by the described procedure [16,17] using *P. falciparum* strain 3D7 sensitive to chloroquine (the LDH method). The samples were made with concentrations of 50, 10, 5, 1, 0.5, 0.1, 0.05, and 0.01 μg/mL. The parasites used in this test were synchronized (Ring stage) with ± 0.3% parasitemia (2% hematocrit). A total of 1 µL of the test solution with various concentrations was taken in to each well (well 96) and then added 99 µL with parasites (each concentration repeated three times). The well plates were put in the chamber with mix gas atmosphere (O_2_ 5%, CO_2_ 5% and N_2_ 90%) and incubated for 72 h at 37 °C. After that, the plate was harvested and stored at −30 °C. After 24 h, the *p*LDH assay was performed by reading the absorbance at 650 nm using a SpectraMax Paradigm^®^ Multi-Mode microplate reader.

### 2.7. In Silico Molecular Docking Studies and ADMET Prediction

The 2-D structures of compounds **1**–**5** were drawn using ChemDraw 18.0 and converted to the 3-D. Their minimum energy was calculated using Chem 3D 18.0 and then stored as mol2. {SYBYL2 (*. Mol2)} format. The crystal structure of human lysosomal acid-alpha-glucosidase (PDB ID: 5NN8) and *P. falciparum* lactate dehydrogenase enzyme (PDB ID: 1CET) with the ligands acarbose and chloroquine was retrieved from the Protein Data Bank. The docking results were expressed as mol dock score (MDS), the energy required in the ligand-receptor interaction, and based on these values, the antidiabetic or antiplasmodial activity of the compounds were predicted. The ADMET properties of the most active compound were calculated by using ProTox Online Tool (http://scistore.cambridgesoft.com, accessed on 5 May 2021), Open babel GUI 2.4.1 (https://sourceforge.net, accessed 25 May 2021), Toxtree version 2.6.6 (http://toxtree.sourceforge.net, accessed on 13 June 2021), pkCSM (http://biosig.unimelb.edu.au/pkcsm, accessed on 21 June 2021), and preADMET (http://preadmet.bmdrs.kr, accessed on 3 July 2021).

### 2.8. Statistical Analysis

The study was conducted three times to determine the mean value (mean SD). A linear regression equation was generated for determination of the percentage of α-glucosidase and α-amylase inhibition concentration of each compound. The difference measured to be statistically significant when the *p*-value < 0.05.

## 3. Results and Discussion

### 3.1. Structure Elucidation

The phytochemical investigation of CH_2_Cl_2_ extract of the bark of *G. forbesii* produced five xanthones namely lichexanthone (**1**) [18,19], subelliptenone H (**2**) [20], 12b-hydroxy-des-D-garcigerrin A (**3**) [21,22], garciniaxanthone B (**4**) [23] and garcigerin A (**5**) [24]. All isolated compounds were elucidated by spectroscopic methods including NMR and HRESIMS analysis, and literature data comparison. To the best of our knowledge, this is the first report of the known compounds **1**–**5** from *G. forbesii* (Figure 1).

### 3.2. Antioxidant Activity

The antioxidant activity depends on the existence of hydroxyl groups and substituents on the aromatic ring. The antioxidant activity of phenolic compounds is well known to be related to the presence of *ortho*- and *para*-substituted hydroxyl groups on the aromatic ring and carbonyl groups as well [25]. The antioxidant activity of isolated xanthones **1**–**5** was assayed by DPPH, ABTS and FRAP methods. DPPH is known for its stability and free radicals scavenging activity [26]. A radical species containing nitrogen atoms of ABTS stabilizes the free radicals through proton donors [27]. The ABTS assay can be applied to lipophilic and hydrophilic compounds [28]. Furthermore, the iron reductive properties of compounds which is an important part of antioxidant activity can be evaluated by using FRAP method [29].

The free radical scavenging power based on DPPH test with quercetin and gallic acid as standards indicated that the inhibitory concentration 50% (IC_50_) of compounds **2**–**5** was ranging from 14.1 to 20.4 μM at a concentration of 159.7 μg/mL (Table 3). Compound **1** was inactive with inhibition below 50%. Previously reported garcinoxanthones SV, garcinone E, and 1,3,6,7-tetrahydroxyxanthone from *G. mangostana* L. showed significant DPPH scavenging capacity with IC_50_ values of 68.55, 63.05, and 28.45 μM, respectively, compared to ascorbic acid (IC_50_ = 48.03 μM). The hydroxyl position of groups gave significant impact on the antioxidant activity of the latter compound [30]. The ABTS antioxidant activity of compounds **2**–**5** was significant with IC_50_ values ranging from 0.05–7.9 μM (Table 3). Compound **1** showed no significant activity with low inhibition at 99 μg/mL (<50%). In particular, compound **3** was found to be the most potent with IC_50_ value 4-fold lower than quercetin. Other compounds with xanthone skeleton such as α-mangostin and γ-mangostin isolated from *G. mangostana* L. showed high activity in ABTS radical scavenging [31]. In addition, the antioxidant evaluation of compounds **1**–**5** using FRAP test, as shown in Table 3, demonstrated that the reduction power of compound **3** was found to be the highest one with value of 203.9 ± 1.19 µM Fe^2+^/g. This activity was followed by compounds **4**, **5**, and **2** with values of 192.7, 187.8, and 166.3 µM Fe^2+^/g, respectively. The iron reducing power of compound **3** was approximately 6-fold greater than ascorbic acid as standard (30.6 µM Fe^2+^/g). These results implied that compounds **2**–**5** are potent antioxidant compounds. The reducing power of other xanthones with carbonyl group and halogen substituents, i.e., (*R*)-6-chloro-2-{[(1-hydroxypropan-2-yl)(methyl)amino]-methyl}-9H-xanthen-9-one hydrochloride and (*R*,*S*)-2-chloro-7-{[(1-hydroxybutan-2-yl)amino]methyl}-9H-xanthen-9-one synthesis results are reported better than vitamin C [32].

The antioxidant evaluation of isolated compounds **1**–**5** argued that the presence of dimethylallyl moiety and additional hydroxy groups on the structures of **2**–**5** was assumed to play important roles in enhancing the antioxidant capacity. Notably, the lack of pyran ring in **3** gave a potent ABTS scavenging activity with IC_50_ value lower than the standards quercetin and gallic acid.

### 3.3. In Vitro Antidiabetic Assay

The antidiabetic potential of compounds **1**–**5** was evaluated on α-glucosidase and α-amylase enzymes. The α-glucosidase inhibitory activity of the five tested compounds was performed using sucrose and maltose as the substrates, and acarbose as the positive control [33]. The α-glucosidase is known to break the 1,4-α bond of carbohydrate into its monomers such as glucose and fructose [34]. The results of α-glucosidase assay, as shown in Table 4, showed that compound **5** was the most potent compound with IC_50_ value of 37.4 μM followed by compounds **2**, **4**, and **3** using sucrose as substrate. In contrast, only compounds **2** and **5** inhibited the enzyme activity with IC_50_ values lower than 100 μM using maltose as substrate. Compound **1** was inactive in both substrates. A new xanthone namely subelliptenone F had a significant effect on α-glucosidase with an IC_50_ value of 4.1 μM compared to acarbose (IC_50_ = 900.0 μM) [9].

α-Amylase is one of the key enzymes responsible for degrading starch to glucose in the human body [35,36]. The inhibitory activity of the tested compounds on α-amylase enzyme indicated that all compounds, except compound **1**, were active with IC_50_ values of 10.8–41.3 μM. The highest activity was shown by compound **4** with IC_50_ value 10.8 μM. It is noted that compounds **2**–**5** are active as α-glucosidase and α-amylase inhibitors. This activity was supposed to be influenced by the dimethylallyl group and pyran rings in the xanthone structure. Compound **1** without dimethylallyl group and pyran rings was not active. The antidiabetic trend of compounds **1**–**5** was similar with the antioxidant inhibition pattern.

### 3.4. In Vitro Antiplasmodial Assay

The in vitro antiplasmodial activity of compounds **1**–**5** against *P. falciparum* strain 3D7 was assessed by lactate dehydrogenase of *P. falciparum* (pLDH) assay [37]. Table 5 displayed that Compounds **2** and **3** were active against *P. falciparum* strain 3D7 with inhibition percentage of 86.8 ± 0.3% (IC_50_ 3.3 ± 0.04 μM) and 87.9 ± 0.2% (IC_50_ 5.0 ± 0.04 μM) at a concentration of 10 μg/mL. Compounds **1**, **4**, and **5** were not active due to their inhibitory activity below 10%. The standard chloroquine showed 98.8 ± 0.25% inhibition and IC_50_ value of 0.006 ± 0.01 μM. These results implied that compounds **2** and **3** are potent as antimalarial agents against the *P. falciparum* 3D7 strain [38]. New xanthones Mckeanianones A, B, C, D, E from *G. mckeaniana* possessing a pyrano ring and two isoprene units also exhibited antimalarial activity against *P. falciparum* strains, TM4 and K1, and cytotoxic activity against the Vero cell line [7].

### 3.5. Molecular Docking Studies

#### 3.5.1. Human Lysosomal Acid-α-glucosidase Enzyme (PDB ID: 5NN8)

α-Glucosidase is an essential enzyme for sugar metabolism in human body [39]. In this study, the molecular docking of isolated xanthones **2** and **5** was studied against human lysosomal acid-alpha-glucosidase enzyme (PDB ID: 5NN8) by Molegro Virtual Docker 5.5 compared to acarbose. Interaction analysis indicated that acarbose associates through hydrogen bonding with the residues Arg404(2); Arg600; Asp616(2); Ser676(2); Ser679; Nya674; Leu678; Gly651 and steric interactions at residue sites of Asp616(3); Asp518; Leu678(3); Trp481(2); Trp618(2); Phe649; Gly651.

The docking results indicated that the total energy and MDS of compound **2** were 108.388 kcal/mol and −78.81 kcal/mol, and 144.003 kcal/mol and −74.54 kcal/mol for Compound **5** (Table 6). The binding mode of these compounds, as shown in Table 6 and Figure 2, revealed that compounds **2** and **5** inhibit the acid-alpha-glucosidase through hydrogen bonding and steric interaction. These compounds are bound to the active site of human lysosomal acid-alpha-glucosidase enzyme (5NN8) through hydrogen-bonding interaction of carbonyl, chelate hydroxyl, and free hydroxyl groups with Arg600, Asp518, and Asp404 residues. The steric interaction of compound **2** was formed by the oxygen atom of a carbonyl group with Met519 residue, xanthone backbone and the pyran ring with Asp282(2) and Phe649 residues, and the dimethylallyl group with Trp376 residue. The similar interaction was also formed by compound **5** with Asp282 (2), Asp518, Asp616, Trp613 (2), Trp516, Trp376, Phe649, Met519, and His674 residues.

#### 3.5.2. *Plasmodium falciparum* Lactate Dehydrogenase Enzyme (PDB ID: 1CET)

The molecular docking of Xanthones **2** and **3** was also studied on *P. falciparum* lactate dehydrogenase enzyme (PDB ID: 1CET). The total energy and MDS of compound **2**, as shown in Table 7, was found to be 108.388 kcal/mol and −103.69 kcal/mol. Compound **3** showed values of 79.747 kcal/mol and −86.38 kcal/mol for the total energy and MDS.

The interaction analysis indicated that compound **2** bound to 1CET via hydrogen bonding of oxygen atom of hydroxyl chelate group (C1) with Tyr85 residue and oxygen atom of carbonyl with Phe52 residue. The steric interaction was established by xanthone backbone and the dimethylallyl group of compound **2** with Tyr85, Glu122(2), Asp53(2), Gly27, Ala98, and Phe100 residues. The hydrogen bonding was also formed by carbonyl group, chelate hydroxyl, and free hydroxyl (C5) of compound **3** with Tyr85(2) and Phe52 residues. The Glu122, Tyr85, Asp53(2), and Ile119 residues of 1CET bound to carbonyl groups, free hydroxyl (C5), and dimethylallyl group of compound **3** by steric interactions (Figure 3).

#### 3.5.3. ADMET Profiles

The absorption, distribution, metabolism, excretion, and toxicity properties of isolated compounds **1**–**5**, as derived from ProTox online tool, revealed that all compounds had molecular weight less than 500 which is important for penetrability [40]. The Caco-2 permeability for prediction of orally administered drug absorption of compounds **1**, **3**, and **4** was > 0.90 and, 0.90 for compounds **2** and **5** (Table 8). This implied that compounds **1**, **3**, and **4** had high permeability in contrast to compounds **2** and **5** [41]. Compounds **1**–**5** showed high intestinal absorption (80–98%) and would be absorbed in the small intestine [42]. The transdermal efficacy as illustrated by skin permeability of compounds **1**–**5** was ranging from −2.735 to −2.851 cm/hour (<−2.5) which mean that they will penetrate the skin properly. It is known that molecules will show difficulty in the skin penetration if the logKp value were higher than −2.5 cm/hour [43]. The circulation in blood plasma (VDss) came out to be acceptable for all compounds **1**–**5** with value higher than −0.15 [44]. An important parameter for reducing side effects and toxicity represented by penetration via blood-brain barrier (BBB) showed that compounds **2** and **4** will sufficiently be able to penetrate, but compounds **2**, **3**, and **5** were very low to reach the brain. The CNS permeability signified for the permeability of the blood surface to the central nerve defined that compounds **1**–**4** with values of −2.89 to −1.679 better than acarbose and chloroquine, were able to permeate a central nervous system [45]. Table 8 informed that isolated xanthones, except **4** were noninhibitors for CYP2D6 and for CYP3A4 and will not interfere the CYP450 biotransformation in general, while compound **4** will be metabolized [46]. In terms of excretion, the total clearance of all compounds shared positive values meaning that they will be discharged quickly except for **5**. The adverse interactions of compounds **1**–**5** with OCT2 inhibitors as denoted by OCT2 substrate parameter revealed that compounds **1**–**5** showed no potential contraindication. The toxicity level of compounds **1**–**5** predicted by using pkCSM-pharmacokinetics for hepatotoxicity and acute oral toxicity in rats (LD_50_) represented the value was in the range 1.889 to 2.057 lower than chloroquine and acarbose with LD_50_ value 2.888 and 2.495, respectively. Compound **4** had the highest LD_50,_ of 2.057 and were the most toxic than others [47]. Hepatotoxicity descriptor declared that compounds **4** and **5** cause hepatotoxicity while compounds **1**–**3** were not hepatotoxic.

## 4. Conclusions

In this study, five known xanthones **1**–**5** was first reported from *G. forbesii*. The compounds were isolated from the CH_2_Cl_2_ extract, and their structure were determined by spectroscopic analysis and literature data comparison. The antioxidant evaluation indicated compounds **3** had the highest reducing activity (203.9 ± 1.19 µM Fe^2+^/g) and ABTS assay (IC_50_ 0.05 ± 0.01 µM) among others. Compound **5** had the highest α-glucosidase inhibition activity with IC_50_ values of 37.4 ± 1.20 μM using sucrose and 91.0 ± 1.14 μM using maltose as substrates. The IC_50_ value of compound **4** against α-amylase was 10.8 ± 0.04 μM. The antiplasmodial assay revealed that compounds **2** and **3** were active against *P. falciparum* strain 3D7 with IC_50_ 3.3 ± 0.04 μM and 5.0 ± 0.04 μM. The *in silico* molecular docking of selected xanthones was studied against human lysosomal acid-alpha-glucosidase (5NN8) and *P. falciparum* lactate dehydrogenase enzymes (1CET). Results from ADMET prediction analyzed using ProTox online tool showed most of compounds generally display good absorption, distribution, metabolism, excretion, and toxicity properties.

## Figures and Tables

**Figure 1 biomedicines-09-01380-f001:**
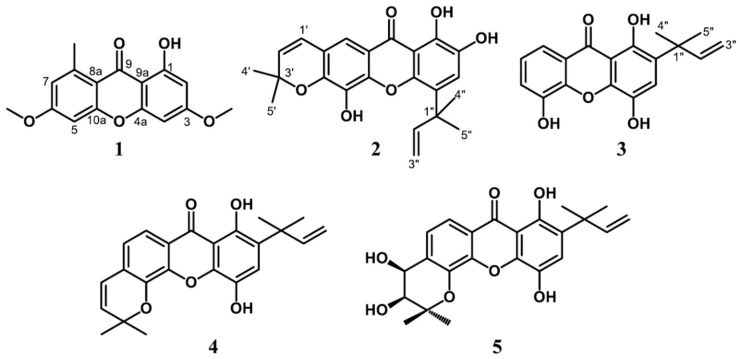
The isolated compounds (**1**–**5**) from the stem barks of *G. forbesii*.

**Figure 2 biomedicines-09-01380-f002:**
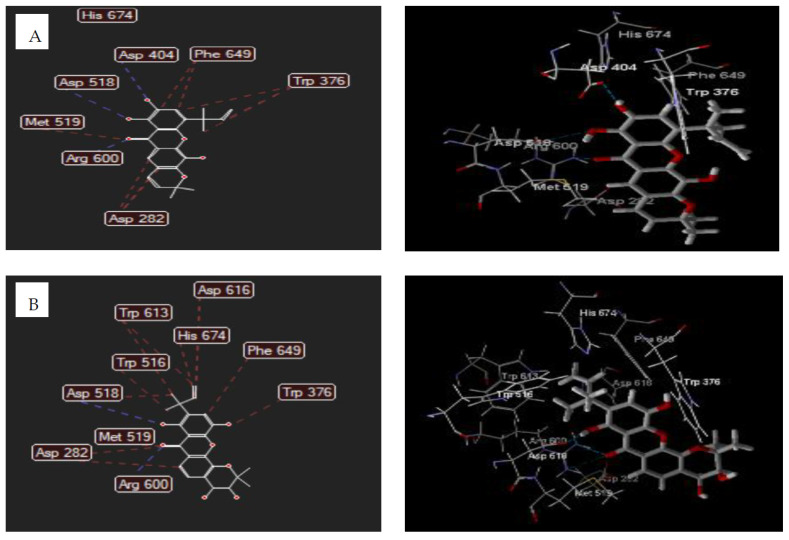
The 2D and 3D interaction of **2** (**A**) and **5** (**B**) with human lysosomal acid-alpha-glucosidase enzyme. Hydrogen bonds were depicted as blue dotted lines while steric interactions were shown as red lines. Carbon atoms were represented in gray, oxygen in red and hydrogen in white.

**Figure 3 biomedicines-09-01380-f003:**
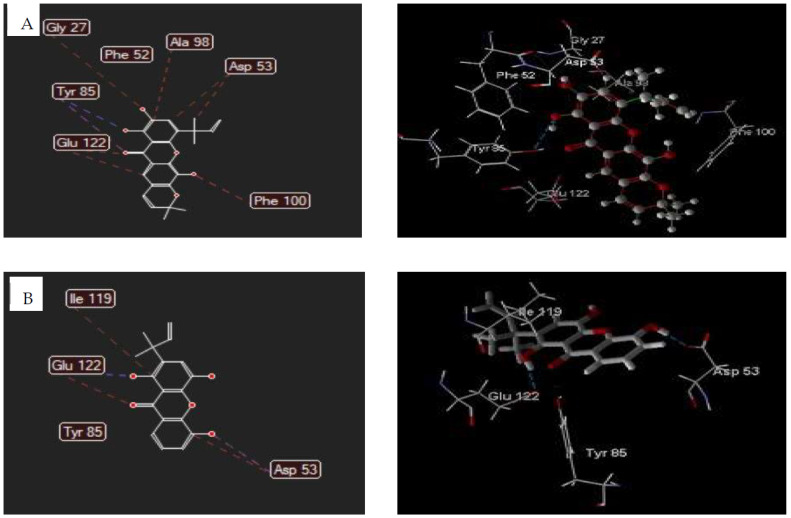
The 2D and 3D interaction of **2** (**A**) and **3** (**B**) with *P. falciparum* lactate dehydrogenase. Hydrogen bonds were depicted as blue dotted lines while steric interactions were shown as red lines. Carbon atoms were represented in gray, oxygen in red and hydrogen in white.

**Table 1 biomedicines-09-01380-t001:** ^1^H (400 MHz) and ^13^C (100 MHz) NMR of Compounds **1**–**3** (in ppm).

Position	1 (CDCl_3_)		2 (DMSO-*d_6_*)		3 (DMSO-*d_6_*)	
	δ_H_ (*J* in Hz)	δ_C_	δ_H_ (*J* in Hz)	δ_C_	δ_H_ (*J* in Hz)	δ_C_
1		163.9		146.6		151.0
2	6.30, d (2.3)	96.8		143.4		127.4
3		165.9	7.26, s	125.4	7.28, s	122.0
4	6.32, d (2.3)	92.2		118.8		136.2
4a		157.1		133.7		141.2
10a		159.5		146.4		144.6
5	6.65, d (2.5)	98.6		121.6		146.2
6		163.8		139.4	7.34, d (7.0)	120.8
7	6.68, d (2.5)	115.6		112.4	7.30, t (7.0)	124.2
8		143.6	7.41, s	113.8	7.60, d (7.0)	114.8
8a		128.0		114.8		120.5
9		182.5		182.4		182.7
9a		104.2		109.1		108.1
1′			5.91, d (10.0)	122.6		
2′			6.59, d (10.0)	132.5		
3′				79.5		
4′			1.58, s	28.4		
5′′			1.58, s	28.4		
1′′				110.7		98.4
2′′			6.31, dd (17.5, 10.6)	147.1	6.23, dd (17.8, 10.3)	146.6
3′′			5.10, d (17.5) 5.01, d (10.6)	111.2	5.01, d (17.8) 4.97, d (10.3)	110.7
4′′			1.46, s	27.4	1.47, s	26.3
5′′			1.46, s	27.4	1.47, s	26.3
1-OH	13.69, s		12.84, s		12.75, s	
2-OH			9.41, s			
4-OH					10.15, s	
5-OH			9.16, s		9.30, s	
3-OMe	4.20 s	55.8				
6-OMe	4.18 s	55.9				
8-CH_3_	3.15, s	23.6				

Note: s: singlet; d: douplet; t: triplet; dd: doublet of doublet.

**Table 2 biomedicines-09-01380-t002:** ^1^H (400 MHz) and ^13^C (100 MHz) NMR of Compounds **4**–**5** (in ppm).

Position	4 (DMSO-*d_6_*)		5 (DMSO-*d_6_*)	
	δ_H_ (*J* in Hz)	δ_C_	δ_H_ (*J* in Hz)	δ_C_
1		151.1		151.2
2		126.6		122.7
3	7.35, s	122.6	7.33, s	122.5
4		136.3		136.3
4a		140.6		141.8
10a		144.9		141.1
5		141.9		147.5
6		127.3		137.9
7	7.20, d (8.0)	121.3	7.37, d (8.0)	120.6
8	7.66, d (8.0)	116.4	7.69, d (8.0)	117.0
8a		121.7		108.6
9		182.3		182.5
9a		108.5		120.7
1′	6.05, d (10.0)	120.2	5.38, d (4.7)	71.8
2′	6.60, d (10.0)	134.5	4.36, d (4.7)	98.3
3′		77.7		69.7
4′	1.48, s	27.7	1.22, s	25.8
5′	1.48, s	27.7	1.20, s	25.4
1′′		99.5		85.7
2′′	6.25, dd (17.9, 10.2)	146.7	6.21, dd (17.6, 10.5)	146.7
3′′	5.01, d (17.9) 4.99, d (10.2)	110.8	4.98, d (17.6) 4.98, d (10.5)	110.8
4′′	1.58, s	26.3	1.45, s	26.3
5′′	1.58, s	26.3	1.45, s	26.3
1-OH	12.91, s		12,87, s	
4-OH	9.49, s		9.57, s	
1′-OH			6.00, s	
2′-OH			4.73, s	

Note: s: singlet; d: douplet; t: triplet; dd: doublet of doublet.

**Table 3 biomedicines-09-01380-t003:** Antioxidant activity of compounds **1**–**5**.

Compounds	Antioxidant Activity				
	FRAP	DPPH		ABTS	
	µM Fe^2+^/g	% Inhibition (159.7 μg/mL)	IC_50_ (μM)	% Inhibition (99 μg/mL)	IC_50_ (μM)
**1**	5.7 ± 0.57	41.60 ± 0.02	>1000	33.24 ± 0.01	>1000
**2**	166.3 ± 1.47	95.29 ± 0.05	19.4 ± 0.15	97.29 ± 0.04	2.7 ± 0.05
**3**	203.9 ± 1.19	96.83 ± 0.03	18.9 ± 0.10	98.31 ± 0.05	0.05 ± 0.01
**4**	192.7 ± 0.77	95.38 ± 0.08	14.1 ± 0.07	96.15 ± 0.01	7.9 ± 0.01
**5**	187.8 ± 1.36	96.60 ± 0.05	20.4 ± 0.22	97.40 ± 0.03	6.5 ± 0.02
Ascorbic acid	30.6 ± 0.27	Nt	Nt	Nt	Nt
Quercetin	Nt	96.34 ± 0.01	4.1 ± 0.01	97.58 ± 0.01	0.17 ± 0.01
Gallic acid	Nt	97.21± 0.01	3.0 ± 0.01	96.30 ± 0.01	0.7 ± 0.01

Note: IC_50_ > 1000 μM = inactive, Nt = not tested.

**Table 4 biomedicines-09-01380-t004:** The in vitro antidiabetic activity of Compounds **1**–**5**.

Compounds	α-Glucosidase		α-Amylase
	Sucrose IC_50_ (μM)	Maltose IC_50_ (μM)	Starch IC_50_ (μM)
**1**	>1000	>1000	>1000
**2**	43.8 ± 1.51	49.3 ± 0.14	41.3 ± 1.24
**3**	79.6 ± 2.01	109.0 ± 1.31	28.7 ± 0.35
**4**	75.9 ± 2.11	139.4 ± 1.21	10.8 ± 0.04
**5**	37.4 ± 1.20	91.0 ± 1.14	18.7 ± 0.54
Acarbose	4.6 ± 0.51	3.4 ± 0.27	4.0 ± 0.32

Note: IC_50_ > 1000 μM = inactive.

**Table 5 biomedicines-09-01380-t005:** The in vitro antiplasmodial activity of compounds **1**–**5**.

Compounds	% Inhibition (10 μg/mL)	IC_50_ (μM)
**1**	˂10	Nt
**2**	86.8 ± 0.33	3.3 ± 0.04
**3**	87.9 ± 0.23	5.0 ± 0.04
**4**	˂10	Nt
**5**	˂10	Nt
Chloroquine	98.8 ± 0.25	0.006 ± 0.01

Note: % inhibition <10 μg/mL = inactive, Nt = not tested.

**Table 6 biomedicines-09-01380-t006:** The total energy, MDS, and interacting residues of **2** and **5** to human lysosomal acid-alpha-glucosidase enzyme (5NN8).

Ligands	Total Energy (kcal/mol)	Mol Dock Score (kcal/mol)	Hydrogen Bonding	Steric Interaction
**2**	108.388	−78.81	Arg600; Asp518; Asp404	Arg600; Phe469(2); Asp518; Trp376
**5**	144.003	−74.54	Arg600(3); Asp518	Asp282 (2); Asp518; Asp616; Trp613 (2); Trp516; Trp376; Phe649; Met519; His674
Acarbose	330.989	−107.31	Arg404(2); Arg600; Asp616(2); Ser676(2); Ser679; His674; Leu678; Gly651	Asp616(3); Asp518; Leu678(3); Trp481(2); Trp618(2); Phe649; Gly651

**Table 7 biomedicines-09-01380-t007:** The total energy, MDS, and interacting residues of **2** and **3** to *P. falciparum* lactate dehydrogenase enzyme (1CET).

Ligand	Total Energy (kcal/mol)	Mol Dock Score (kcal/mol)	Hydrogen Bonding	Steric Interaction
**2**	108.388	−103.69	Tyr85(2); Phe52	Tyr85; Glu122(2); Asp53(2); Gly27; Ala98; Phe100;
**3**	79.747	−86.38	Glu122; Asp53; Tyr85	Glu122; Tyr85; Asp53(2); Ile119
Chloroquine	35.324	−108.99	Gly29; Gly99; Ser28	Gly29; Gly99; Ala98; Asp53

**Table 8 biomedicines-09-01380-t008:** ADMET properties by ProTox Online Tool.

Ligands	Absorption			Distribution			Metabolism		Excretion		Toxicity		
	Caco-2 Permeability	Intestinal Abs	Skin Permeability	VDss	BBB Permeability	CNS Permeability	CYP2D6 Inhibitors	CYP3A4 Inhibitors	Total Clearance	Renal OCT2 Substrate	Oral Rat Acute Toxicity (LD50)	Oral Rat Chronic Toxicity (LOAEL)	Hepato Toxicity
**1**	1.233	95.93	−2.851	0.039	−0.291	−2.098	No	No	0.655	No	2.045	1.449	No
**2**	0.736	98.018	−2.735	0.046	−1.258	−1.93	No	No	0.186	No	1.889	0.696	No
**3**	1.298	96.82	−2.735	0.068	−1.02	−2.029	No	No	0.198	No	2.055	0.633	No
**4**	0.989	94.818	−2.754	0.143	0.099	−1.679	No	Yes	0.176	No	2.057	0.313	Yes
**5**	0.335	80.007	−2.735	−0.014	−1.36	−3.040	No	No	−0.102	No	2.036	1.695	Yes
Chloroquine	1.259	89.440	−2.564	1.757	0.410	−2.687	Yes	No	0.993	Yes	2.888	0.423	Yes
Acarbose	−0.278	0	−2.735	−0.644	−1.854	−7.308	No	No	0.546	No	2.495	7.203	No

## Data Availability

The data presented in this study are available on request from the corresponding author.

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
