# Peer review of "Evaluation of the Antioxidant, Antidiabetic, and Antiplasmodial Activities of Xanthones Isolated from Garcinia forbesii and Their In Silico Studies"

_biomedicines, 2021, doi:10.3390/biomedicines9101380_

Round 1

Reviewer 1 Report

In this manuscript, the authors extracted, for the first time, 5 xanthones from Garcinia forbesii. Characterized them and evaluated their antioxidant, antidiabetic and antiplasmodial activities, as well their in silico studies.

In this paper, authors have done exhaustive and well-documented studies.

I recommend that, in the characterization of the compounds, you check the constants of the protons.

The paper may be acceptable for Biomedicines. 

Author Response

Response to Reviewer 1 Comments

First, I would like to thank you for your constructive comments on our manuscript to improve this script. Here, I want to respond to this comment.

Point 1: I recommend that, in the characterization of the compounds, you check the constants of the protons.

Response 1: I have checked the proton constant and have revised it.

Reviewer 2 Report

The manuscript entitled "Evaluation of the antioxidant, antidiabetic, and antiplasmodial activities of xanthones isolated from Garcinia forbesii and their in silico studies" is well elaborated, the results are clearly presented, and the conclusions are supported by the results. This reviewer recommends the publication of the manuscript in "Biomedicines", but first there are some minor points that should be corrected/taken into consideration by the authors:  

  1. Line 15: Please change the style of "in silico" to italic
  2. Lines 15-17: The sentences "The isolated compounds were evaluated for their antioxidant activity was tested by DPPH, ABTS and FRAP methods. Antidiabetic inhibitory activity was using performed..." are not well constructed. Please rephrase these sentences. 
  3. Lines 16 and 17: The expressions "Antidiabetic inhibitory activity" and "antiplasmodial inhibitory activity" don't make sense. Please replace them by "Antidiabetic activity" and "Antiplasmodial activity", respectively. 
  4. Lines 18: "Molecular docking analysis on the enzyme (5NN8) and (1CET)...". The codes are derived from PDB, however the name of the enzymes is not 5NN8 or 1CET. Please mention the name of the enzyme for each code and the respective PDB ID codes in round brackets. This should be applied throughout the manuscript.  
  5. Line 37: Please replace "benzophenon" by "benzophenone"
  6. Lines 55 and 56: Please replace "there are" by "there is"
  7. Lines 61 and 64: The authors should indicate the extended name of the abbreviations DPPH, ABTS, FRAP, and ADMET when these appear in the manuscript for the first time. This reviewer would recommend using the following order: "extended name (abbreviation)" (for example: 2,2-diphenyl-1-picrylhydrazyl (DPPH)) and apply this for all abbreviations mentioned for the first time. 
  8. Line 193: "The crystal structure of (...) antiplasmodial (PDB ID: 1CET)...". Please mention the name of the enzyme corresponding to the code 1CET. 
  9. Line 203: Please replace "determination the" by "determination of the". 
  10. Line 207: Please replace the title "Struktur elucidation" by "Structure elucidation"
  11. Line 211: Please replace the word "dan" by "and". 
  12. Line 237: Please replace the word "positio" by "position"
  13. Line 292: Please replace the word "This" by "These"
  14. Line 312: Please indicate the name of the enzyme corresponding to the code 5NN8. 
  15. Line 376: Please replace the word "was" by "were"

Author Response

Response to Reviewer 2 Comments

First of all, I would like to thank you for your constructive comments on our manuscript to improve this manuscript. Here, I would like to respond to those comments.

Point 1: Line 15: Please change the style of "in silico" to italic

Response 1: This has been revised

Point 2: Lines 15-17: The sentences "The isolated compounds were evaluated for their antioxidant activity was tested by DPPH, ABTS and FRAP methods. Antidiabetic inhibitory activity was using performed..." are not well constructed. Please rephrase these sentences. 

Response 2: This has been revised and added

Point 3: Lines 16 and 17: The expressions "Antidiabetic inhibitory activity" and "antiplasmodial inhibitory activity" don't make sense. Please replace them by "Antidiabetic activity" and "Antiplasmodial activity", respectively. 

Response 3: This has been revised

Point 4: Lines 18: "Molecular docking analysis on the enzyme (5NN8) and (1CET)...". The codes are derived from PDB, however the name of the enzymes is not 5NN8 or 1CET. Please mention the name of the enzyme for each code and the respective PDB ID codes in round brackets. This should be applied throughout the manuscript.  

Response 4: This has been revised and added

Point 5: Line 37: Please replace "benzophenon" by "benzophenone"

Response 5: This has been revised

 Point 6: Lines 55 and 56: Please replace "there are" by "there is"

 Response 6: This has been revised

Point 7: Lines 61 and 64: The authors should indicate the extended name of the abbreviations DPPH, ABTS, FRAP, and ADMET when these appear in the manuscript for the first time. This reviewer would recommend using the following order: "extended name (abbreviation)" (for example: 2,2-diphenyl-1-picrylhydrazyl (DPPH)) and apply this for all abbreviations mentioned for the first time. 

Response 7: This has been revised and added

Point 8: Line 193: "The crystal structure of (...) antiplasmodial (PDB ID: 1CET)...". Please mention the name of the enzyme corresponding to the code 1CET. 

Response 8: This has been revised and added

Point 9: Line 203: Please replace "determination the" by "determination of the". 

Response 9: This has been revised

Point 10: Line 207: Please replace the title "Struktur elucidation" by "Structure elucidation"

Response 10: This has been revised

Point 11: Line 211: Please replace the word "dan" by "and".

Response 11: This has been revised

Point 12: Line 237: Please replace the word "positio" by "position"

Response 12: This has been revised

 Point 13: Line 292: Please replace the word "This" by "These"

Response 13: This has been revised

Point 14: Line 312: Please indicate the name of the enzyme corresponding to the code 5NN8. 

Response 14: This has been revised and added

Point 15: Line 376: Please replace the word "was" by "were"

Response 15: This has been revised